# Improved Synthesis of Phosphoramidite-Protected *N*^6^-Methyladenosine via BOP-Mediated S_N_Ar Reaction

**DOI:** 10.3390/molecules26010147

**Published:** 2020-12-31

**Authors:** Shifali Shishodia, Christopher J. Schofield

**Affiliations:** 1Department of Chemistry, Chemistry Research Laboratory, University of Oxford, 12 Mansfield Road, Oxford OX1 3TA, UK; sshishodia@mcw.edu; 2Department of Biochemistry, Medical College of Wisconsin, Milwaukee, WI 53226, USA

**Keywords:** *N*^6^-methyladenosine, BOP-mediated S_N_Ar, ethanoadenosine, phosphoramidite, RNA epigenetics

## Abstract

*N*^6^-methyladenosine(m^6^A) is the most abundant modification in mRNA. Studies on proteins that introduce and bind m^6^A require the efficient synthesis of oligonucleotides containing m^6^A. We report an improved five-step synthesis of the m^6^A phosphoramidite starting from inosine, utilising a 1-*H*-benzotriazol-1-yloxytris(dimethylamino)phosphoniumhexafluorophosphate (BOP)_-_mediated S_N_Ar reaction in the key step. The route manifests a substantial increase in overall yield compared to reported routes, and is useful for the synthesis of phosphoramidites of other adenosine derivatives, such as ethanoadenosine, an RNA analogue of the DNA adduct formed by the important anticancer drug Carmustine.

## 1. Introduction

Modifications to mRNA are presently of interest from basic biological science and medicinal chemistry perspectives [1]. *N*^6^-methyladenosine(m^6^A) is the most abundant modification to mRNA, being reported to be present at an average of 2–3 sites/animal mRNA molecule [2]. Some of the methyltransferases, demethylases and reader domains that introduce, remove and bind m^6^A in mRNA, respectively, are linked to disease [3]. Studies on the roles of such enzymes/proteins and the medicinal chemistry targeting them requires the efficient synthesis of oligonucleotides containing m^6^A. Research on m^6^A has been hampered by its high cost, due to low yields for the reported seven to eight-step syntheses of phosphoramidite-protected m^6^A and the use of relatively expensive starting materials [4,5,6,7]. In one strategy, acetylation of the hydroxyls of inosine is followed by Vilsmeier-type aryl chlorination giving 2′,3′,5′-*O*-triacetyl-6-chloroinosine [8] (Scheme 1A). S_N_Ar reaction with methylamine and cleavage of the acetyl groups gives m^6^A (**1**, Scheme 1A) which can be phosphoramidite protected. Alternatively, the hydroxyls of adenosine are first silyl protected using Beigelman’s strategy [9], then the exocyclic amine is protected by benzoylation or acetylation. Methylation is achieved using methyliodide, however, formation of *N*^1^-methyladenosine(m^1^A) significantly compromises the m^6^A yield [4] (Scheme 1B).

Here we describe a five-step synthesis of the m^6^A phosphoramidite from inosine employing BOP-mediated S_N_Ar reaction. The lack of use of POCl_3_ or methyliodide enables a two to three-fold improvement in the m^6^A phosphoramidite yield compared to reported methods (Scheme 2) [4,5,6,7]. BOP- or PyBOP-mediated S_N_Ar reaction is also useful for the synthesis of m^6^A analogues [10,11], e.g., methylethanoadenosine and ethylethanoadenosine.

## 2. Results

### 2.1. Synthesis of m^6^A Phosphoramidite

In our synthesis of the m^6^A phosphoramidite, the 5′- and 3′-hydroxyls of inosine were first selectively protected using di-*tert*-butylsilyl bis(trifluoromethanesulfonate) (DTBS ditriflate) (Scheme 3). The 2′-hydroxyl group was then protected using *tert*-butyldimethylsilyl chloride (TBDMSCl) in a one-pot procedure [9] (Beigelman’s strategy, Scheme 3). The resultant 3′,5′-*O*-(di-*tert*-butyl)silyl-2′-*O*-dimethyl(*tert*-butyl)silyl-inosine (**2**) was treated with BOP using 1,8-diazabicyclo [5.4.0]undec-7-ene (DBU) as a base, to give *O*^6^-benzotriazol-3′,5′-*O*-(di-*tert*-butyl)silyl-2′-*O*-dimethyl(*tert*-butyl)silyl-inosine (II, Figure 1), which was reacted in situ with excess methylamine to give 3′,5′-*O*-(di-*tert*-butyl)silyl-2′-*O*-dimethyl(*tert*-butyl)silyl-*N*^6^-methyladenosine [10] (**3**, 98% apparent yield, over two steps, Scheme 3). The DTBS group of **3** was selectively cleaved using a dilute solution of HF.pyridine, to give 2′-*O*-dimethyl(*tert*-butyl)silyl-*N*^6^-methyladenosine [4] (**4**, 90%, Scheme 3). The 5′-hydroxyl of **4** was selectively protected using 4,4′-dimethoxytritylchloride (DMTrCl) to give 5′-*O*-(4,4′-dimethoxytrityl)-2′-*O*-dimethyl(*tert*-butyl)silyl-*N*^6^-methyladenosine (**5**, 85%, Scheme 3) as reported [7]; portionwise addition of DMTrCl gave an improved yield. Note that TBDMS group migration from the 3′- to the 2′-hydroxyl and vice-versa occurs in protic solvents [12], hence use of such solvents, e.g., methanol, should be avoided for chromatography of **5**.

Phosphitylation of the 3′-hydroxyl of **5** was achieved using 2-cyanoethyl-*N*,*N*-diisopropylchlorophosphoramidite to give 5′-*O*-(4,4′-dimethoxytrityl)-(3′-*O*-[(2-cyanoethyl)(*N,N*-diisopropylamino)phosphino]-2′-*O*-dimethyl(*tert*-butyl)silyl-*N*^6^-methyladenosine (**6**, 80%, Scheme 3), using 10 equivalents of base and 2.5 equivalents of the phosphitylating reagent with moisture-free purification under Ar. The optimised route comprises a five-step procedure proceeding to a 60% overall yield from inosine, a substantial advantage over reported routes (Scheme 2) [4,5,6,7]. 

We also investigated the direct reaction of BOP-activated inosine with methylamine to give m^6^A (**1**, 90%, Scheme 4); treatment of m^6^A (**1**) with DMTrCl gave 5′-*O*-(4,4′-dimethoxytrityl)-*N*^6^-methyladenosine (**7**) as the major crude product (Scheme 4). Treatment of the crude product mixture with TBDMSCl/imidazole, gave a mixture of the 2′- and 3′- TBDMS regioisomers of protected m^6^A (**8** and **9**, Scheme 4). However, this mixture was difficult to separate by flash column chromatography; a low yield (28–37%) of the desired isomer (**8**, Scheme 4) has been reported in the literature [7,8]. Thus, although this procedure is shorter, it is presently less efficient than our afore-described route. 

### 2.2. Synthesis of m^6^A Analogues

To test the generality of the S_N_Ar reaction, we explored the synthesis of m^6^A analogues (**11a–g**, Scheme 5). Thus, treatment of inosine with BOP/DBU in DMF (40 min) was followed by addition of an amine and reaction overnight to give the desired products in generally good non-optimised yields. 

### 2.3. Synthesis of N^1^,N^6^-Ethanoadenosine and N^6^, N^1^,N^6^-(m)Ethylethanodenosine Phosphoramidites

We also investigated the synthesis of *N*^1^,*N*^6^-ethanoadenosine and analogues using the BOP-mediated S_N_Ar reaction. Following treatment of 2′,3′,5′-*O*-tris[dimethyl(*tert*-butyl)silyl]inosine (**12**, Scheme 6) with BOP/DBU, addition of excess ethanolamine, methylethanolamine, or ethylethanolamine resulted in the desired 6-substituted adenosine derivatives (**13a**–**c**, Scheme 6) in high yields (86–90%). 2′,3′,5′-*O*-Tris[dimethyl(*tert*-butyl)silyl]-*N*^6^-(2-hydroxyethyl)adenosine (**13a**, Scheme 6) was treated with (triphenoxy)phosphoniummethyl iodide to give intermediate 2′,3′,5′-*O*-tris[dimethyl(*tert*-butyl)silyl]-*N*^6^-(2-iodoethyl)adenosine (**14a**, Scheme 6) which spontaneously cyclises to give the desired 2′,3′,5′-*O*-tris[dimethyl(*tert*-butyl)silyl]-*N*^1^,*N*^6^-ethanoadenosine (**15a**, 60%, Scheme 6). Cyclisation of **13** (**b** and **c**) using (triphenoxy)phosphonium-methyl iodide gives the positively charged cyclic products (**15b** (50%) and **15c** (36%), Scheme 6). TBDMS deprotection using 3HF.Et_3_N [13] (**15a**–**c**, Scheme 6) gave *N*^1^,*N*^6^-ethanoadenosine, *N*^6^,*N*^1^,*N*^6^-methylethanoadenosine, or *N*^6^,*N*^1^,*N*^6^-ethylethanoadenosine (**16a**–**c**, 22–70%, Scheme 6). 

## 3. Discussion

Overall, we have developed a modular and efficient five-step synthesis of phosphoramidite-protected m^6^A starting from inosine, which avoids the use of POCl_3_/the Vilsmeier reagent (Scheme 1A). The key step comprises the efficient (90% yield) BOP-mediated S_N_Ar reaction of methylamine with a readily prepared protected inosine derivative, Although, there are reports of using BOP-mediated S_N_Ar to prepare ribonucleoside analogues [10,14,15,16,17], use of the reaction to prepare phosphoramidite-protected materials suitable for oligonucleotide synthesis has been limited [18,19]. Investigations on the mechanism of BOP-mediated S_N_Ar reactions are reported [11]. Thus, the oxide formed by base-mediated deprotonation of the hydroxyl group on the aromatic inosine ring reacts with the electrophilic phosphorus of BOP to form an (acyloxy)phosphonium intermediate (I, Figure 1), which undergoes S_N_Ar reaction with the oxybenzotriazole to form an intermediate (II**,**
Figure 1) which is subsequently replaced by a nucleophile in a second S_N_Ar reaction to form the desired product (III, Figure 1). The method is also suited for the synthesis of phosphoramidites of other purines/pyrimidines modified at the C-6/C-4 positions, respectively, as shown by the preparation of *N*^1^,*N*^6^-ethanoadenosine and its analogues using BOP-mediated S_N_Ar (Scheme 6); these reactions resulted in moderate (22%) to very good (90%) non-optimised yields ( (**11b**, Scheme 5) (in the case of *N*^6^-ethyladenosine, the yield was low due to a non-optimal work-up procedure), showing the generality of the reaction. The method also works with alcohol nucleophiles, though since MeOH is a poor nucleophile, after the formation of the proposed oxybenzotriazol intermediate (II, Figure 1), the crude mixture was evaporated and redissolved in excess methanol in the presence of a base (DBU) to give the desired *O*^6^-methylinosine (**11d**) product (72%) (Scheme 5). The absence of protection of the hydroxyl groups of inosine does not substantially impact on yields as evident from the formation of **11(e–g)** vs. **13(a–c)**, although, more polar solvent is required to dissolve unprotected inosine.

The modified nucleosides described here will have use in ongoing investigations to probe the selectivity and inhibition of RNA modifying enzymes, including m^6^A demethylases and m^6^A reader proteins. The BOP-mediated S_N_Ar reaction may also have other applications, e.g., in improving the reported route to *N*^1^,*N*^6^-ethanodeoxyadenosine phosphoramidite [20] or because *N*^1^,*N*^6^-ethanoadenosine can be oxidised to *N*^1^,*N*^6^-ethenoadenosine using MnO_2_, to give a fluorescent analogue of adenosine [21].

## 4. Materials and Methods 

### 4.1. General Experimental Considerations

Reagents for synthesis were from Sigma-Aldrich, Alfa Aesar, Cambridge Biotech, Fischer Scientific, or Link Technology, unless otherwise stated. Anhydrous solvents used in reactions were either analytical grade, as obtained commercially (Alfa Aesar), or were freshly distilled. HPLC grade solvents were employed for work-up and chromatography. For the chromatographic purification of phosphoramidites, solvents were dried over P_2_O_5_ prior to use. Reactions involving moisture-sensitive reagents were carried out under an argon atmosphere; glassware was oven dried and cooled under nitrogen before use. Reagents were used as supplied (analytical or HPLC grade) without prior purification. Anhydrous MgSO_4_ was used as a drying agent.

Thin layer chromatography was performed using aluminium plates coated with 60 F254 silica. Plates were visualised using UV light (254 nm), or 1% (*m/v*) aq. KMnO_4_ stain. Flash column chromatography was performed using Kieselgel 60 silica in a glass column, or on a Biotage SP4 flash column chromatography platform. Retention factors (R_f_) are quoted to a precision of 0.05.

Deuterated solvents were from Sigma and Apollo Scientific Ltd. ^1^H-NMR and ^13^C-NMR spectra were recorded using Bruker AVIII400, AVII500, AVIII600 and AVIII700 NMR spectrometers (Bruker, Banner Lane, UK). Fields were locked by external referencing to the relevant residual deuterium resonance. Chemical shifts (δ) are reported in ppm; coupling constants (*J*) are recorded in Hz to the nearest 0.5 Hz; when peak multiplicities are reported, the following abbreviations are used: s = singlet, d = doublet, t = triplet, q = quartet, m = multiplet, br = broadened, dd = doublet of doublets, dt = doublet of triplets, td = triplet of doublets. Spectra were recorded at room temperature unless otherwise stated. ^1^H and ^13^C-NMR spectra of compounds are available in the online Appendix A.

Low-resolution mass spectra (*m*/*z*) and high-resolution mass spectra (HRMS) were recorded using an LCT Premier XE (Waters, Elstree, UK) or a microTOF machine (Bruker, Banner Lane, UK).

Melting points were recorded on a Gallenkamp Hot Stage apparatus (Gallenkamp, Loughborough, UK). IR spectra were recorded using a Bruker Tensor 27 FT-IR spectrometer (Bruker, Banner Lane, UK)as thin films. Selected characteristic peaks are reported in cm^−1^.

### 4.2. Experimental Details for Synthesis

#### General Procedure A: Synthesis of *N*-Alkyladenosine 

DBU (1.5 mmol) was added dropwise to a stirred solution of inosine (1 mmol) and BOP (1.2 mmol) in DMF; the mixture was then heated at 40 °C. After the consumption of starting material (approximately 40 min, as assessed by TLC), the reaction was cooled to room temperature and the appropriate amine (5 mmol) was added dropwise and the reaction was stirred overnight. The crude product mixture was concentrated under reduced pressure, then diluted with ethyl acetate and was washed with water (3 × 10 mL). The organic layer was dried (anhydrous MgSO_4_) and concentrated under vacuum. The resulted solid was recrystallised twice from *iso*-propanol.

***N*^6^-Methyladenosine** (**1**). The desired product was prepared according to General Procedure A; white solid (250 mg, 90%). m.p 214.5 °C; ^1^H-NMR (500 MHz, D_2_O) δ 2.93 (s, 3H), 3.74 (dd, *J* = 13.0, 3.5 Hz, 1H), 3.83 (dd, *J* = 13.0, 2.5 Hz, 1H), 4.20 (q, *J* = 3.0 Hz, 1H), 4.32 (dd, *J* = 5.0, 3.0 Hz, 1H), 4.66 (t, *J* = 5.5 Hz, 1H), 5.90 (d, *J* = 6.0 Hz, 1H), 8.02 (s, 1H), 8.11 (s, 1H); ^13^C-NMR (126 MHz, D_2_O) δ 27.5, 62.1, 71.1, 74.0, 88.36, 88.39, 120.0, 140.1, 152.9, 155.6. HRMS (ESI) *m*/*z*: calculated for C_11_H_16_O_4_N_5_ [M + H]^+^ 282.1197, observed: 282.1196.

**3′,5′-*O*-(Di-*tert*-butyl)silyl-2′-*O*-dimethyl(*tert*-butyl)silylinosine** (**2**). The desired compound was prepared according to a modified version of the reported procedure [22]. To a stirred suspension of inosine (2.12 g, 8 mmol) in 40 mL anhydrous DMF at 0 °C, di-*t*-butylsilyl ditrifluoromethanesulfonate (3.0 mL, 8.8 mmol) was added dropwise under an N_2_ atmosphere. After consumption of starting material (30 min, as assessed by TLC), the reaction was quenched immediately with imidazole (2.7 g, 40 mmol) at 0 °C. After 5 min, the reaction was warmed to room temperature. *t*-Butyldimethylsilyl chloride (1.5 g, 9.6 mmol) was then added portionwise and the reaction was refluxed at 60 °C for 12 h. The suspension was then cooled to room temperature, water was added, and the precipitate was collected by suction filtration. The filtrate was discarded, and the white precipitate was washed with cold methanol. The methanol layer was evaporated under reduced pressure and the product was crystallised from CH_2_Cl_2_ to give a white solid (4.0 g, 98%). m.p 191–193.4 °C. TLC R_f_ 0.45 (3:2 cyclohexane/ethyl acetate); ^1^H NMR (600 MHz, CDCl_3_) δ 0.17 (s, 3H), 0.18 (s, 3H), 0.96 (s, 9H), 1.07 (s, 9H), 1.10 (s, 9H), 4.02–4.09 (m, 1H), 4.25 (td, *J* = 10.0, 5.0 Hz, 1H), 4.38 (dd, *J* = 9.5, 4.5 Hz, 1H), 4.45–4.58 (m, 2H), 5.96 (s, 1H), 7.87 (s, 1H), 8.11 (s, 1H), 12.56 (s, 1H); ^13^C NMR (151 MHz, CDCl_3_) δ −5.0, −4.3, 18.3, 20.4, 22.8, 25.9, 27.0, 27.5, 67.8, 74.8, 75.89, 75.94, 92.3, 125.5, 138.3, 144.7, 148.1, 158.9; HRMS (ESI) *m*/*z*: calculated for C_24_H_43_O_4_N_5_^28^Si_2_ [M + H]^+^ 523.2767, observed: 523.2756.

**3′,5′-*O-*Bis(*tert*-butyl)silyl-2′-*O*-(*tert*-butyldimethyl)silyl-*N*^6^-methyladenosine** (**3**). The desired compound was prepared according to a modified version of the reported procedure [10]. To a stirred solution of 3′,5′-*O-*Bis(*tert*-butylsilyl)-2′-*O*-(*tert*-butyldimethylsilyl)inosine (**2**; 663 mg, 1.2 mmol) and BOP (0.64 g, 1.44 mmol) in 20 mL of THF, DBU (0.3 mL, 1.8 mmol) was added dropwise and the mixture was heated at 40 °C. After the consumption of the starting material (40 min, as assessed by TLC), the reaction was cooled to room temperature and methylamine (0.3 mL, 6.0 mmol) was added dropwise and the reaction was stirred overnight. The crude product mixture was concentrated under reduced pressure and diluted with ethyl acetate and was washed with water (3 × 10 mL). The organic layer was dried (anhydrous MgSO_4_) and concentrated under vacuum. The residue was purified by column chromatography (9:1 to 3:2 cyclohexane/ethyl acetate) which resulted in an oil (665 mg, 98%). TLC R_f_ 0.20 (7:3 cyclohexane/ethyl acetate); ^1^H NMR (400 MHz, CDCl_3_) δ 0.00 (s, 3H) 0.02 (s, 3H) 0.78 (s, 9H) 0.90 (s, 9H) 0.94 (s, 9H) 3.05 (d, *J* = 1.0 Hz, 3H) 3.86–3.90 (m, 1H) 4.02–4.10 (m, 1H) 4.34 (dd, *J* = 9.0, 5.0 Hz, 1 H) 4.38–4.44 (m, 1 H) 4.47 (d, *J* = 4.5 Hz, 1 H) 5.76 (b.s, 2 H) 7.62 (s, 1 H) 8.22 (s, 1 H); ^13^C NMR (101 MHz, CDCl_3_) δ −5.0, −4.3, 18.3, 20.4, 22.8, 25.9, 27.1, 27.5, 27.6, 67.9, 74.6, 75.5, 75.8, 92.4, 120.5, 125.0, 138.0, 153.4, 155.5; HRMS (ESI) *m*/*z*: calculated for C_25_H_46_O_4_N_5_^28^Si_2_ [M + H]^+^ 536.3082, observed: 536.3078. Analytical data are consistent with those reported [4].

**2′-*O*-(*tert*-Butyldimethyl)silyl-*N*^6^-methyladenosine** (**4**). The desired compound was prepared according to the reported procedure [4]. To a stirred solution of 3′,5′-*O-*Bis(*tert*-butylsilyl)-2′-*O*-(*tert*-butyldimethylsilyl)-*N*^6^-methyladenosine (**3**; 240 mg, 0.45 mmol) in 4 mL of CH_2_Cl_2_ at −15 °C, a cooled solution of (HF)_x_·pyridine (0.06 mL, 2.3 mmol) in 365 μL pyridine was added. The reaction temperature was maintained at -15 °C and stirred for 12 h. The reaction was diluted with CH_2_Cl_2_, then washed first with sat. aq. NaHCO_3_ solution, then with water (3 × 10 mL). The organic layer was dried (anhydrous MgSO_4_) and concentrated under reduced pressure. The residue was purified by column chromatography (9:1 to 3:2 cyclohexane/ethyl acetate) which resulted in oil (160 mg, 90%). TLC R_f_ 0.15 (2:3 hexane/ethyl acetate); ^1^H-NMR (400 MHz, CDCl_3_) δ 0.00 (s, 3H), 0.02 (s, 3H), 0.94 (s, 9H), 3.42 (d, *J* = 1.0 Hz, 3H), 3.89 (dd, *J* = 10.5, 9.0 Hz, 1H), 4.01–4.11 (m, 1H), 4.34 (dd, *J* = 9.0, 5.0 Hz, 1H), 4.41 (dd, *J* = 9.0, 5.0 Hz, 1H), 4.47 (d, *J* = 5.0 Hz, 1H), 5.76 (s, 2H), 7.62 (s, 1H), 8.22 (s, 1H); ^13^C NMR (101 MHz, CDCl_3_) δ −5.4, −5.3, 17.9, 25.6, 25.8, 27.5, 63.5, 73.1, 74.4, 87.8, 91.3, 119.7, 140.0, 140.1, 152.9, 155.8; HRMS (ESI) *m*/*z*: calculated for C_17_H_30_O_4_N_5_^28^Si [M + H]^+^ 396.2062, observed: 396.2068. Analytical data are consistent with those reported [4].

**5′-*O*-(4,4′-Dimethoxytrityl)−2′-*O*-dimethyl(*tert*-butyl)silyl-*N*6-methyladenosine (5**). The desired compound was prepared according to the reported procedure [4]. To a stirred solution of 2′-*O*-dimethyl(*tert*-butyl)silyl-*N*6-methyladenosine (**4**) (2.6 g, 6.6 mmol) in 4 mL anhydrous pyridine at 0 °C, DMTrCl (2.7 g, 8.0 mmol) was added portionwise at regular intervals for 12 h. The reaction was quenched by addition of an excess of anhydrous methanol (0.5 mL) at room temperature. After 1 h, the solution was concentrated under vacuum. The crude solid was first dissolved and fractioned between aqueous NaHCO_3_ and ethyl acetate; the organic layer was then washed with water (3 × 10 mL). The organic layer was dried (MgSO_4_) and concentrated under vacuum. The residue was purified by column chromatography (9:1 to 3:2 cyclohexane/ethyl acetate) resulted in a green oil (3.9 g, 85%). TLC R_f_ 0.45 (2:3 cyclohexane/ethyl acetate); ^1^H-NMR (400 MHz, CDCl_3_) δ −0.13 (s, 3H) 0.00 (s, 3H) 0.86 (s, 9H) 2.77 (d, *J* = 4.0 Hz, 1H) 3.17 (s, 3H) 3.36–3.43 (m, 1H) 3.54 (dd, *J* = 10.5, 3.5 Hz, 1H) 3.80 (s, 6 H) 4.27 (d, *J* = 3.5 Hz, 1H) 4.33–4.37 (m, 1H) 5.02 (t, *J* = 5.5 Hz, 1H) 5.85 (d, *J* = 4.5 Hz, 1H) 6.04 (b.s, 2H) 6.83 (d, *J* = 9.0 Hz, 4H) 7.18–7.28 (m, 3H) 7.36 (d, *J* = 8.0 Hz, 4H) 7.47 (dd, *J* = 8.5 Hz, 1.5, 2H) 7.98 (s, 1H) 8.35 (s, 1H); ^13^C NMR (101 MHz, CDCl_3_) δ −5.6, −5.5, 18.3, 25.8, 25.9, 55.2, 60.4, 63.0, 73.6, 75.5, 85.0, 87.5, 89.2, 113.4, 120.0, 127.3, 128.1, 128.3, 130.39, 130.45, 135.9, 138.0, 145.0, 153.0, 155.4, 158.89, 158.91; HRMS (ESI) *m*/*z*: calculated for C_38_H_48_O_6_N_5_^28^Si [M + H]^+^ 698.3368, observed: 698.3359. Analytical data are consistent with those reported [4].

5′-*O*-(4,4′-Dimethoxytrityl)-(3′-*O*-[(2cyanoethyl)(*N*,*N*-diisopropylamino)phosphino]−2′-*O*-dimethyl(*tert*-butyl)silyl-*N*^6^-methyladenosine (6). The desired compound was prepared according to the reported procedure [4]. To a stirred solution of 5′-*O*-(4,4′-dimethoxytrityl)−2′-*O*-dimethyl(*tert*-butyl)silyl-*N*^6^-methyladenosine (5, 500 mg, 0.7 mmol) in anhydrous CH_2_Cl_2_ in an over-dried flask under argon, DIPEA (1.3 mL, 7.2 mmol) was added dropwise and the reaction mixture was allowed to stir at 0 °C for 10 min. (2-Cyanoethyl)-*N,N*-diisopropylchlorophosphoramidite (0.40 mL, 1.8 mmol) was added to the reaction mixture dropwise at 0 °C under an argon atmosphere. The reaction was stirred at 0 °C for 30 min, then gradually (about 30 min) warmed to room temperature. After another five hours under an inert atmosphere, the reaction mixture was treated with a saturated aq. KCl solution, then evaporated by rotary evaporation. The desired product was separated by silica gel column chromatography (1:1:0.01 hexane/ethyl acetate/pyridine) resulting in a colourless oil (520 mg, 80%) yield. TLC R_f_ 0.40 (1:1:0.01 hexane/ethyl acetate/pyridine); ^1^H NMR (700 MHz, CD_2_Cl_2_-*d_2_*) Major peaks are listed. δ −0.15 (s, 3H), −0.01 (s, 3H), 0.82 (s, 9H), 1.10 (s,3H), 1.11 (s, 3H), 1.22 (s, 3H), 1.22 (s, 3H), 1.65 (s, 2H), 2.62–2.74 (m, 2H), 3.19 (s, 3H), 3.36 (dd, *J* = 10.5, 4.5 Hz, 1H), 3.54 (dd, *J* = 10.5, 4.0 Hz, 1H), 3.82 (s, 6H), 3.85–3.93 (m, 1H), 3.95–4.10 (m, 1H), 4.41–4.49 (m, 1H), 5.12 (dd, *J* = 6.1, 4.4 Hz, 1H), 5.33–5.40 (m, 2H), 5.79 (s, 1H), 5.99 (d, *J* = 6.0 Hz, 1H), 6.78–6.90 (m, 4H), 7.23–7.29 (m, 1H), 7.28–7.34 (m, 2H), 7.34–7.40 (m, 4H), 7.47–7.52 (m, 2H), 7.94 (s, 1H), 8.25 (s, 1H); ^13^C NMR (176 MHz, CD_2_Cl_2_) Major peaks are listed. δ −5.4, −5.0, 0.8, 17.8, 20.4, 20.44, 21.1, 24.37, 24.4, 25.4, 25.44, 42.9, 43.0, 55.2, 58.8, 58.9, 63.5, 72.8, 72.9, 74.7, 74.7, 83.46, 83.48, 86.5, 88.4, 113.1, 117.8, 125.2, 126.8, 127.8, 128.1, 128.2, 129.0, 130.10, 130.14, 135.7, 139.0, 144.9, 153.0, 155.5, 158.6, 158.7; ^31^P-NMR (202 MHz, CD_2_Cl_2_) δ 148.0, 150.8.

***N^6^,N*^6^-Dimethyladenosine** (**11a**). The desired product was prepared according to General Procedure A; white solid (250 mg, 85%). ^1^H NMR (600 MHz, D_2_O) δ 3.06 (s, 6H), 3.75 (dd, *J* = 13.0, 3.5 Hz, 1H), 3.85 (dd, *J* = 13.0, 2.5 Hz, 1H), 4.19 (q, *J* = 3.0 Hz, 1H), 4.29–4.33 (t, *J* = 4.5 Hz, 1H), 4.59 (t, *J* = 5.5 Hz, 1H), 5.82 (d, *J* = 6.0 Hz, 1H), 7.78 (b.s, 1H), 8.00 (s, 1H); ^13^C-NMR (151 MHz, D_2_O) δ 38.7, 61.4, 70.5, 73.7, 85.5, 88.2, 118.9, 138.2, 148.2, 151.3, 153.6; HRMS (ESI) *m*/*z*: calculated for C_12_H_18_O_4_N_5_ [M + H]^+^ 296.1353, observed: 296.1352.

***N*^6^-Ethyladenosine** (**11b**). The desired product was prepared according to General Procedure A; Ethylamine was prepared in-situ. To a stirred solution of ethylamine hydrochloride (1.1 g, 13.7 mmol) in 10 mL of ethanol in 50 mL round bottom flask, Ag_2_O (3.8 g, 16.4 mmol) was added and the mixture was stirred at room temperature under N_2_ for 1 h. The precipitate was collected by suction filtration; the resultant solution was then added to the stirred mixture of inosine, BOP and DBU. Yellowish solid (90 mg, 30%). ^1^H NMR (600 MHz, D_2_O + DMSO-*d_6_*) δ 2.63 (t, *J* = 2.0 Hz, 3H), 3.53 (b.s, 2H), 3.77 (dd, *J* = 13.0, 3.5 Hz, 1H), 3.85 (dd, *J* = 13.0, 3.0 Hz, 1H), 4.23 (q, *J* = 3.0 Hz, 1H), 4.35 (dd, *J* = 5.0, 3.0 Hz, 1H), 5.98 (d, *J* = 6.5 Hz, 1H), 8.18 (s, 1H), 8.24 (s, 1H), (1H under solvent peak); ^13^C-NMR (151 MHz, D_2_O + DMSO-*d_6_*) δ 13.9, 61.6, 70.8, 73.7, 86.0, 88.3, 117.6, 130.0, 140.1, 152.7, 154.6, (1C under solvent peak); HRMS (ESI) *m*/*z*: calculated for C_12_H_18_O_4_N_5_ [M + H]^+^ 296.1308, observed: 296.1353.

***N^6^-*Cyclopropyladenosine** (**11c**). The desired product was prepared according to General Procedure A; white solid (222 mg, 72%). ^1^H-NMR (700 MHz, DMSO-*d_6_* + D_2_O) δ 0.62–0.71 (m, 2H), 0.88 (dd, *J* = 7.0, 2.0 Hz, 2H), 3.00 (bs, 1H), 3.67 (dd, *J* = 12.0, 3.5 Hz, 1H), 3.77 (dd, *J* = 12.0, 3.5 Hz, 1H), 4.21–4.28 (m, 1H), 4.67 (t, *J* = 5.5 Hz, 1H), 5.97 (d, *J* = 6.0 Hz, 1H), 8.32 (s, 1H), 8.41 (s, 1H), (1 proton under solvent); ^13^C NMR (176 MHz, DMSO-*d_6_* + D_2_O) δ 7.0, 61.9, 70.8, 70.9, 73.9, 86.2, 86.3, 88.4, 119.8, 140.3, 140.4, 152.7, 155.9; HRMS (ESI) *m*/*z*: calculated for C_13_H_18_O_4_N_5_ [M + H]^+^ 308.1353, observed: 308.1351.

***O*^6^-Methylinosine** (**11d**). DBU (1.5 mmol) was added dropwise to a stirred solution of inosine (1 mmol), BOP (1.2 mmol) in THF; the mixture was heated at 40 °C. After the consumption of starting material (40 min, as assessed by TLC), the reaction mixture was concentrated under reduced pressure and an excess of MeOH was added to the flask and the reaction was stirred at 40 °C overnight. The crude product mixture was concentrated under reduced pressure and diluted with ethyl acetate, then washed with water (3 × 10 mL). The organic layer was dried (MgSO_4_) and concentrated under vacuum. The crude mixture was purified (99:1 to 9:1 ethyl acetate/methanol) by column chromatography which resulted in a white solid (0.2 g, 72%). TLC R_f_ 0.3 (9:1 CH_2_Cl_2_/MeOH); ^1^H-NMR (600 MHz, DMSO-*d_6_*) δ 3.58 (ddd, *J* = 12.0, 6.0, 4.0 Hz, 1H), 3.69 (dt, *J* = 12.0, 4.5 Hz, 1H), 3.98 (q, *J* = 4.0 Hz, 1H), 4.11 (s, 3H), 4.17 (q, *J* = 4.5 Hz, 1H), 4.60 (q, *J* = 5.5 Hz, 1H), 5.13 (t, *J* = 5.5 Hz, 1H), 5.22 (d, *J* = 5.0 Hz, 1H), 5.50 (d, *J* = 6.0 Hz, 1H), 6.00 (d, *J* = 5.5 Hz, 1H), 8.57 (s, 1H), 8.63 (s, 1H); ^13^C-NMR (151 MHz, DMSO-*d_6_*) δ 54.5, 61.8, 70.8, 74.2, 86.2, 88.2, 121.6, 142.9, 152.2, 152.24, 160.9; calculated for C_11_H_13_O_5_N_4_ [M + H]^+^ 283.0934, observed: 283.0932.

***N*^6^-(2-Hydroxyethyl)adenosine** (**11e**). The desired product was prepared according to General Procedure A; white solid (190 mg, 61%). ^1^H-NMR (600 MHz, DMSO-*d*_6_) δ 3.57 (dd, *J* = 12.0, 4.0 Hz, 1H), 3.67 (dd, *J* = 12.0, 3.5 Hz, 1H), 4.13–4.21 (m, 1H), 4.51–4.64 (m, 3H), 5.96 (d, *J* = 6.0 Hz, 1H), 8.49 (s, 1H), 8.52 (s, 1H) (4 protons under the residual water peak); ^13^C NMR (151 MHz, DMSO-*d_6_*) δ 61.6, 63.4, 70.7, 74.0, 86.1, 88.2, 121.5, 142.7, 152.0, 152.2, 160.6; HRMS (ESI) *m*/*z*: calculated for C_12_H_18_O_5_N_5_ [M + H]^+^312.1302, observed: 312.1297.

***N*^6^,*N*^6^-Methyl(2-hydroxyethyl)adenosine** (**11f**). The desired product was prepared according to General Procedure A; white solid (293 mg, 90%). ^1^H-NMR (700 MHz, DMSO-*d_6_*) δ 3.56 (ddd, *J* = 12.0, 7.0, 3.5 Hz, 1H), 3.61–4.43 (m, 6H), 4.59 (q, *J* = 6.0 Hz, 1H), 4.75 (t, *J* = 5.5 Hz, 1H), 5.18 (d, *J* = 5.0 Hz, 1H), 5.37 (dd, *J* = 7.0, 4.5 Hz, 1H), 5.45 (d, *J* = 6.0 Hz, 1H), 5.91 (d, *J* = 6.0 Hz, 1H), 8.22 (s, 1H), 8.37 (s, 1H) (3 methyl protons and one hydroxyl group under the residual water peak in DMSO); ^13^C NMR (176 MHz, DMSO-*d_6_*) δ 37.3, 52.7, 60.0, 62.0, 71.0, 73.9, 86.2, 88.3, 120.1, 139.2, 150.4, 152.2, 154.6; HRMS (ESI) *m*/*z*: calculated for C_13_H_20_O_5_N_5_ [M + H]^+^ 326.1459, observed: 326.1459.

***N*^6^,*N*^6^-Ethyl(2-hydroxyethyl)adenosine** (**11g**). The desired product was prepared according to General Procedure A; white solid (275 mg, 85%). ^1^H-NMR (700 MHz, D_2_O) δ 1.18 (t, *J* = 7.0 Hz, 3H), 2.53 (d, *J* = 9 Hz, 1H)3.75 (dd, *J* = 13.0, 3.5 Hz, 1H), 3.78–4.09 (m, 7H), 4.21 (q, *J* = 3.5 Hz, 1H), 4.34 (dd, *J* = 5.0, 3.5 Hz, 1H), 5.98 (d, *J* = 6.0 Hz, 1H), 8.13 (s, 1H), 8.17 (s, 1H) ^13^C-NMR (176 MHz, D_2_O) δ 15.4, 44.6, 59.6, 61.1, 61.5, 70.6, 73.5, 85.8, 88.1, 119.3, 138.7, 149.2, 152.0, 154.1; HRMS (ESI) *m*/*z*: calculated for C_14_H_22_O_5_N_5_ [M + H]^+^ 340.1617, observed: 340.1615.

**2′,3′,5′-*O*-Tris(*tert*-butyldimethyl)silylinosine** (**12**). To a stirred solution of inosine (3.75 g, 13.24 mmol) and imidazole (3.6 g, 53.0 mmol) in anhydrous DMF in a 50 mL round bottom flask, TBDMSCl (6.6 g, 43.7 mmol) was added portionwise. The reaction was heated at 60 °C for 12 h. The suspension was cooled to room temperature, water was added and the precipitate was collected by suction filtration. The filtrate was discarded, and the white precipitate was washed with cold methanol. The methanol layer was evaporated under vacuum; the product was crystallised as a white solid from CH_2_C1_2_ (7.8 g, 94%). TLC R_f_ 0.6 (1:9 MeOH/CH_2_Cl_2_); ^1^H-NMR (400 MHz, CDCl_3_) δ −0.31 (s, 3H), −0.16 (s, 3H), −0.04 (s, 3H), −0.03 (s, 3H), 0.00 (s, 3H), 0.01 (s, 3H), 0.68 (s, 9H), 0.79 (s, 9H), 0.82 (s, 9H), 3.66 (dd, *J* = 11.5, 2.5 Hz, 1H), 3.86 (dd, *J* = 11.5, 4.0 Hz, 1H), 4.00 (q, *J* = 3.5 Hz, 1H), 4.17 (t, *J* = 4.0 Hz, 1H), 4.38 (t, *J* = 4.5 Hz, 1H), 5.88 (d, *J* = 5.0 Hz, 1H), 7.97 (s, 1H), 8.09 (s, 1H), 12.83 (s, 1H); ^13^C NMR (101 MHz, CDCl_3_) δ −5.4, −5.0, −4.70, −4.66, −4.4, 17.9, 18.1, 18.6, 25.7, 25.8, 26.1, 62.4, 71.8, 85.5, 88.3, 125.0, 139.1, 144.5, 148.9. 159.2; HRMS (ESI) *m/z*: calculated for C_28_H_55_O_5_N_4_^28^Si_3_ [M + H]^+^ 611.3474, observed: 611.3468.

**2′,3′,5′-*O*-Tris[dimethyl(*tert*-butyl)silyl]-*N*^6^-(2-hydroxyethyl)adenosine** (**13a**). To a stirred solution of 2′,3′,5′-*O*-tris(*tert*-butyldimethyl)silylinosine (**12**; 0.1 g, 0.16 mmol) and PyBOP (0.1 g, 0.2 mmol) in 10 mL of THF in a 50 mL round bottom flask, DIPEA (42 μL, 0.24 mmol) was added dropwise and the mixture was heated at 40 °C. After the consumption of the starting material (40 min, as assessed by TLC), the reaction was cooled to room temperature and ethanolamine (0.2 mL, 0.35 mmol) was added dropwise; the reaction was then stirred overnight. The crude product mixture was concentrated under reduced pressure and then diluted with ethyl acetate and was washed with water (3 × 10 mL). The organic layer was dried (MgSO_4_) and concentrated under reduced pressure. The residue was purified by column chromatography (99:1 to 94:6 CH_2_Cl_2_/MeOH) which resulted in oil (95 mg, 90%). ^1^H NMR (500 MHz, CD_3_OD) δ −0.28 (s, 3H), −0.02 (s, 3H), 0.17 (s, 6H), 0.18 (s, 6H), 0.79 (s, 9H), 0.98 (s, 9H), 0.99 (s, 9H), 3.71–3.84 (m, 4H), 3.85 (dd, *J* = 11.5, 3.0 Hz, 1H), 4.06 (dd, *J* = 11.0, 4.5 Hz, 1H), 4.15 (dt, *J* = 5.0, 3.0 Hz, 1H), 4.40 (dd, *J* = 4.5, 2.5 Hz, 1H), 4.83 (dd, *J* = 6.0, 4.5 Hz, 1H), 4.89 (s, 5H), 6.07 (d, *J* = 6.0 Hz, 1H), 8.26 (s, 1H), 8.31 (s, 1H), (-OH peak under solvent peak); ^13^C-NMR (126 MHz, CD_3_OD) δ −6.6, −6.5, −6.3, −5.6, −5.6, −5.5, 17.4, 17.6, 18.0, 24.9, 25.1, 25.2, 45.97, 46.0, 60.4, 62.7, 72.7, 76.0, 86.2, 87.7, 119.5, 139.3, 148.6, 152.5, 155.0; HRMS (ESI) *m*/*z*: calculated for C_30_H_60_O_5_N_5_^28^Si_3_ [M + H]^+^ 635.3713, observed: 635.3797.

**2′,3′,5′-*O*-Tris[dimethyl(*tert*-butyl)silyl]-*N*^6^,*N*^6^-methyl(2-hydroxyethyl)adenosine** (**13b**). To a stirred solution of 2′,3′,5′-*O*-tris(*tert*-butyldimethyl)silylinosine (**12**; 1 g, 1.64 mmol) and BOP (0.9 g, 1.96 mmol) in 25 mL of EtOH in a 50 mL round bottom flask, DBU (0.3 mL, 1.97 mmol) was added dropwise; the mixture was heated at 40 °C. After the consumption of the starting material (40 min, TLC), the reaction was cooled to room temperature and methylethanolamine (0.65 mL, 8.2 mmol) was added dropwise; the reaction was then stirred overnight. The crude product mixture was concentrated under reduced pressure, then diluted with ethyl acetate and was washed with water (3 × 10 mL). The organic layer was dried (anhydrous MgSO_4_) and concentrated under reduced pressure. The residue was purified by column chromatography (99:1 to 94:6 CH_2_Cl_2_/MeOH) which resulted in oil (0.82 g, 80%). TLC R_f_ 0.6 (6:94 MeOH/CH_2_Cl_2_). ^1^H-NMR (600 MHz, CDCl_3_) δ −0.29 (s, 3H), −0.14 (s, 3H), −0.01 (s, 3H), 0.00 (s, 3H), 0.02 (s, 3H), 0.03 (s, 3H), 0.71 (s, 9H), 0.83 (s, 9H), 0.85 (s, 9H), 3.42 (s, 3H), 3.67 (dd, *J* = 11.5, 3.0 Hz, 1H), 3.87 (t, *J* = 5.0 Hz, 2H), 3.92 (dd, *J* = 11.4, 4.0 Hz, 1H), 3.94–4.09 (m, 4H), 4.21 (t, *J* = 4.0 Hz, 1H), 4.58 (t, *J* = 4.5 Hz, 1H), 5.92 (d, *J* = 5.0 Hz, 1H), 7.97 (s, 1H), 8.20 (s, 1H); ^13^C-NMR (151 MHz, CDCl_3_) δ −5.38, −5.36, −5.0, −4.72, −4.70, −4.4, 17.9, 18.1, 18.5, 25.7, 25.9, 26.1, 53.8, 61.6, 62.5, 71.9, 75.6, 85.3, 88.3, 120.4, 137.7, 150.5, 152.2, 155.6; HRMS (ESI) *m*/*z*: calculated for C_31_H_62_O_5_N_5_^28^Si_3_ [M + H]^+^ 668.4053, observed: 668.4042.

**2′,3′,5′-*O*-Tris[dimethyl(*tert*-butyl)silyl]-*N*^6^,*N*^6^-ethyl(2-hydroxyethyl)adenosine** (**13c**). To a stirred solution of 2′,3′,5′-*O*-tris(*tert*-butyldimethyl)silylinosine (**12**; 1 g, 1.64 mmol) and BOP (0.87 g, 1.96 mmol) in 25 mL of EtOH in a 50 mL round bottom flask, DBU (0.3 mL, 2.0 mmol) was added dropwise and the mixture was heated at 40 °C. After the consumption of the starting material (40 min, as assessed by TLC), the reaction was cooled to room temperature and ethylethanolamine (0.65 mL, 8.2 mmol) was added dropwise; the reaction was then stirred overnight. The crude product mixture was concentrated under reduced pressure, then diluted with ethyl acetate and washed with water (3 × 10 mL). The organic layer was dried (anhydrous MgSO_4_) and concentrated under reduced pressure. The residue was purified by column chromatography (99:1 to 94:6 CH_2_Cl_2_/MeOH) which resulted in an oil (960 mg, 86%). TLC R_f_ 0.5 (6:94 MeOH/CH_2_Cl_2_); ^1^H NMR (600 MHz, CDCl_3_) δ −0.30 (s, 3H), −0.15 (s, 3H), −0.02 (s, 3H), −0.01 (s, 3H), 0.00 (s, 3H), 0.01 (s, 3H), 0.69 (s, 9H), 0.82 (s, 9H), 0.83 (s, 9H), 1.18 (t, *J* = 7.0 Hz, 3H), 3.66 (dd, *J* = 11.0, 3.0 Hz, 1H), 3.72–4.15 (m, 8H), 4.21 (t, *J* = 4.0 Hz, 1H), 4.59 (t, *J* = 5.0 Hz, 1H), 4.67–5.17 (m, 1H), 5.89 (d, *J* = 5.0 Hz, 1H), 8.05 (s, 1H), 8.16 (s, 1H); ^13^C NMR (151 MHz, CDCl_3_) δ −5.39, −5.38, −5.0, −4.72, −4.70, −4.4, 13.3, 17.9, 18.1, 18.5, 25.7, 25.8, 26.0, 44.6, 51.6, 62.5, 62.6, 71.9, 75.5, 85.2, 88.3, 120.1, 137.8, 150.5, 152.1, 155.0. HRMS (ESI) *m/z*: calculated for C_32_H_64_O_5_N_5_^28^Si_3_ [M + H]^+^ 682.4209, observed: 682.4201.

**2′,3′,5′-*O*-Tris[dimethyl(*tert*-butyl)silyl]-*N^1^,N^6^*-ethanoadenosine** (**15a**). To a stirred solution of 2′,3′,5′-*O*-tris[dimethyl(*tert*-butyl)silyl]-*N*^1^,*N*^6^-(2-hydroxyethyl)adenosine (**13a**; 1.3 g, 2 mmol) and Et_3_N (1.4 mL, 10 mmol) in 30 mL of anhydrous DMF in a 50 mL round bottom flask, methyltriphenoxyphosphonium iodide (1 g, 2.4 mmol) was added and the mixture was stirred at room temperature for 1 h. Anhydrous methanol was added and the crude product mixture was concentrated under reduced pressure, then diluted with ethyl acetate and washed with NaHCO_3_ and water (3 × 10 mL). The organic layer was dried (MgSO_4_) and concentrated under reduced pressure. The residue was purified by column chromatography (99:1 to 85:15 CH_2_Cl_2/_MeOH) resulting in an oil (0.75 g, 60%). ^1^H-NMR (600 MHz, CDCl_3_) δ −0.18 (s, 3H), −0.07 (s, 3H), 0.00 (s, 3H), 0.01 (s, 3H), 0.05 (s, 3H), 0.06 (s, 3H), 0.75 (s, 9H), 0.83 (s, 9H), 0.87 (s, 9H), 3.71 (dd, *J* = 11.5, 2.5 Hz, 1H), 3.93 (dd, *J* = 11.5, 3.0 Hz, 1H), 4.06 (dt, *J* = 5.5, 3.0 Hz, 1H), 4.18 (t, *J* = 4.5 Hz, 1H), 4.26–4.40 (m, 3H), 4.90–4.95 (m, 2H), 5.92 (d, *J* = 4.0 Hz, 1H), 8.38 (s, 1H), 8.51 (s, 1H); ^13^C NMR (151 MHz, CDCl_3_) δ −5.3, −5.27, −4.8, −4.6, −4.4, −4.2, 17.9, 18.0, 18.5, 25.7, 25.8, 26.1, 26.4, 26.5, 46.28, 46.33, 46.4, 49.1, 62.0, 71.1, 76.7, 85.2, 89.0, 117.6, 142.3, 143.4, 149.0, 151.3; HRMS (ESI) *m*/*z*: calculated for C_30_H_58_O_4_N_5_^28^Si_3_ [M + H]^+^ 636.3767, observed: 636.3791.

***N^1^,N^6^*-Ethanoadenosine** (**16a**). To a stirred solution of 2′,3′,5′-*O*-tris[dimethyl(*tert*-butyl)silyl]-*N^6^*-ethanoadenosine (**15a,** 470 mg, 0.74 mmol) in CH_2_Cl_2_, 3HF.Et_3_N (361 µL, 2.2 mmol) was added dropwise. The solution was left to stir for 48 h. The mixture was reduced under pressure and was purified by column chromatography (99:1 to 9:1 ethyl acetate/methanol) which resulted in a white solid (150 mg, 70%). ^1^H NMR (700 MHz, DMSO-*d*_6_) δ 3.49–3.57 (m, 2H), 3.60–3.68 (m, 1H), 3.86–3.97 (m, 3H), 4.08–4.16 (m, 3H), 4.44 (t, *J* = 5.0 Hz, 1H), 5.11–5.21 (m, 2H), 5.46 (s, 1H), 5.78 (d, *J* = 6.0 Hz, 1H), 8.09 (s, 1H), 8.17 (s, 1H); ^13^C-NMR (176 MHz, DMSO*-d*_6_) δ 46.5, 53.3, 61.9, 70.8, 74.5, 86.1, 88.0, 120.0, 138.5, 145.2, 145.4, 150.2; HRMS (ESI) *m*/*z*: calculated for C_12_H_16_O_4_N_5_ [M + H]^+^ 294.1197, observed: 294.1191.

**2′,3′,5′-*O*-Tris[dimethyl(*tert*-butyl)silyl]-*N^6^,N^1^,N^6^*-methylethanoadenosine** (**15b**). To a stirred solution of 2′,3′,5′-*O*-tris[dimethyl(*tert*-butyl)silyl]-*N*^6^,*N*^6^-methyl(2-hydroxyethyl)adenosine (**13b**; 0.5 g, 0.75 mmol) and Et_3_N (0.54 mL, 3.75 mmol) in 30 mL of anhydrous DMF in a 50 mL round bottom flask, methyltriphenoxyphosphonium iodide (0.85 g, 1.9 mmol) was added and the mixture was stirred at room temperature for 1 h. Anhydrous methanol was added and the crude product mixture was concentrated under reduced pressure, then diluted with ethyl acetate and washed with NaHCO_3_ and water (3 × 10 mL). The organic layer was dried (anhydrous MgSO_4_), then concentrated under reduced pressure. The residue was purified by alumina column chromatography (99:1 to 90:10 CH_3_Cl/MeOH) which resulted in an oil. ^1^H NMR (600 MHz, CDCl_3_) δ −0.20 (s, 3H), −0.12 (s, 3H), −0.08 (s, 3H), −0.07 (s, 3H), 0.00 (s, 3H), 0.01 (s, 3H), 0.70 (s, 9H), 0.75 (s, 9H), 0.81 (s, 9H), 3.57 (s, 3H), 3.65 (dd, *J* = 11.5, 2.0 Hz, 1H), 3.88 (dd, *J* = 11.5, 3.0 Hz, 1H), 4.00 (dt, *J* = 5.0, 2.5 Hz, 1H), 4.12 (dd, *J* = 5.5, 4.0 Hz, 1H), 4.17–4.26 (m, 2H), 4.35–4.42 (m, 1H), 5.05–5.3 (m, 2H), 5.87 (d, *J* = 3.5 Hz, 1H), 8.43 (s, 1H), 8.50 (s, 1H); ^13^C NMR (151 MHz, CDCl_3_) δ −5.4, −5.2, −4.8, −4.6, −4.5, −4.2, 17.9, 18.1, 18.6, 25.7, 25.8, 26.2, 34.6, 45.9, 49.1, 51.6, 61.7, 70.7, 85.0, 89.3, 115.4, 117.0, 129.5, 142.8, 143.7, 149.9, 150.1; HRMS (ESI) *m/z*: calculated for C_31_H_60_O_4_N_5_^28^Si_3_ [M] 650.3947, observed: 650.3924.

**2′,3′,5′-*O*-Tris[dimethyl(*tert*-butyl)silyl]-*N^6^,N^1^,N^6^*-ethylethanoadenosine** (**15c**). To a stirred solution of 2′,3′,5′-*O*-tris[dimethyl(*tert*-butyl)silyl]-*N*^6^,*N*^6^-ethyl(2-hydroxyethyl)adenosine (**15c**; 0.4 g, 0.6 mmol) and Et_3_N ( 0.4 mL, 3 mmol) in 20 mL of anhydrous DMF in a 50 mL round bottom flask, methyltriphenoxyphosphonium iodide (0.6 g, 1.2 mmol) was added; the mixture was stirred at room temperature for 1 h. Anhydrous methanol was added and the crude product mixture was concentrated under reduced pressure and then diluted with ethyl acetate and was washed subsequently with NaHCO_3_ and water (3 × 10 mL). The organic layer was dried (anhydrous MgSO_4_) and concentrated under reduced pressure. The residue was purified by alumina column chromatography (99:1 to 95:5 CH_2_Cl_2_/MeOH) which resulted in a solid (0.26 g, 36%). mp 195 °C; ^1^H-NMR (600 MHz, CDCl_3_) δ −0.25 (s, 3H), −0.18 (s, 3H), −0.03 (s, 3H), −0.01 (s, 3H), 0.00 (s, 3H), 0.01 (s, 3H), 0.70 (s, 9H), 0.74 (s, 9H), 0.83 (s, 9H), 1.33 (t, *J* = 7.0 Hz, 3H), 3.2 (q, *J* = 7.0 Hz, 2H), 3.65 (dd, *J* = 12.0, 2.0 Hz, 1H), 3.88 (dd, *J* = 12.0, 3.0 Hz, 1H), 4.00 (dt, *J* = 5.0, 2.5 Hz, 1H), 4.23–4.28 (m, 2H), 4.59 (t, *J* = 5.0 Hz, 1H), 4.69 (t, *J* = 5.0 Hz, 1H) 5.0–5.18 (m, 2H), 5.83 (d, *J* = 5.0 Hz, 1H), 8.34 (s, 1H), 8.46 (s, 1H); ^13^C-NMR (151 MHz, CDCl_3_) δ −5.3, −5.0, −5.0, −4.6, −4.6, −4.2, 13.0, 17.9, 18.1, 18.5, 25.7, 25.8, 26.2, 35.6, 48.4, 51.6, 61.7, 70.7, 73.4, 85.0, 89.3, 115.4, 144.8, 149.9, 150.0, 152.0; HRMS (ESI) *m*/*z*: calculated for C_31_H_60_O_4_N_5_^28^Si_3_ [M] 664.3886, observed: 664.3896.

***N^6^,N^1^,N^6^*-Methylethanoadenosine** (**16b**). To a stirred solution of 2′,3′,5′-*O*-tris[dimethyl(*tert*-butyl)silyl]-*N*^6^,*N^6^*-methylethanoadenosine (**15b**; 500 mg, 0.8 mmol) in CH_2_Cl_2_, 3HF.Et_3_N (0.4 mL, 2.3 mmol) was added dropwise. The solution was stirred for 48 h. The mixture was reduced under pressure, then purified by alumina column chromatography (99:1 to 9:1 ethyl acetate/methanol) which resulted in a white solid (54 mg, 22%). mp 202.5 °C; ^1^H-NMR (600 MHz, DMSO-*d*_6_) δ 3.01 (s, 3H), 3.29 (m, 1H), 3.38 (dt, *J* = 12.5, 4.5 Hz, 1H), 3.68 (q, *J* = 4.0 Hz, 1H), 3.76–3.84 (m, 2H), 3.87 (b.s, 1H), 4.18 (q, *J* = 5.0 Hz, 1H), 4.43 (t, *J* = 9.5 Hz, 1H), 4.79 (t, *J* = 5.5 Hz, 1H), 5.01 (s, 1H), 5.28–5.37 (m, 2H), 5.68 (d, *J* = 5.0 Hz, 1H), 8.50 (s, 1H), 8.52 (s, 1H); ^13^C-NMR (151 MHz, D_2_O) δ 36.4, 50.5, 53.4, 63.6, 72.6, 77.1, 88.6, 90.8, 119.0, 145.7, 148.0, 152.2, 152.3; HRMS (ESI) *m*/*z*: calculated for C_13_H_18_O_4_N_5_ [M]^+^ 308.1353, observed: 308.1353.

***N^6^,N^1^,N^6^*-Ethylethanoadenosine** (**16c**). To a stirred solution of 2′,3′,5′-*O*-tris[dimethyl(*tert*-butyl)silyl]-*N*^6^,*N^6^*-methylethanoadenosine (**15c**; 470 mg, 0.74 mmol) in CH_2_Cl_2_, 3HF.Et_3_N (360 µL, 2.2 mmol)was added dropwise. The solution was left to stir for 48 h. The mixture was reduced under pressure and was purified by alumina column chromatography (99:1 to 9:1 ethyl acetate/methanol) which resulted in a white solid (65 mg, 30%). mp 210 °C; ^1^H-NMR (600 MHz, DMSO-*d*_6_) δ 1.15 (t, *J* = 7.0 Hz, 3H), 2.88 (q, *J* = 7.0 Hz, 2H), δ 3.49–3.57 (m, 2H), 3.36 (dt, *J* = 12.5 Hz, 4.0 Hz, 1H), 3.54 (q, *J* = 4.0 Hz, 1H), 3.76–3.84 (m, 2H), 3.95 (b.s, 1H), 4.15–4.27 (m, 1H), 4.34–4.43 (m, 1H), 5.1 (b.s, 1H), 5.43–5.57 (m, 2H), 6.0 (d, *J* = 5.0 Hz, 1H), 8.48 (s, 1H), 8.63 (s, 1H); ^13^C-NMR (151 MHz, DMSO-*d*_6_) δ 13.0, 32.4, 48.3, 52.3, 62.4, 70.4, 75.3, 85.4, 90.8, 119.0, 146.4, 148.2, 151.9, 152.0; HRMS (ESI) *m/z*: calculated for C_13_H_18_O_4_N_5_ [M]^+^ 322.1518, observed: 322.1515.

## 5. Conclusions

Overall, we have developed an improved the synthesis of the m^6^A phosphoramidite starting from inosine. Following alcohol group protection, a BOP-mediated S_N_Ar reaction was employed to introduce the desired *N*^6^-methylamino group in a suitably protected form to be efficiently converted to the phosphoramidite for incorporation into oligonucleotides. The BOP-mediated S_N_Ar reaction can be employed to prepare other N^6^-alylamino substituted adenosine derivatives, including ethanoadenosine, an RNA analogue of the DNA adduct formed by the important anticancer drug Carmustine.

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
