# Peer review of "Improved Synthesis of Phosphoramidite-Protected *N*^6^-Methyladenosine via BOP-Mediated S_N_Ar Reaction"

_molecules, 2020, doi:10.3390/molecules26010147_

Round 1
Reviewer 1 Report
The authors described the synthesis of m6A phosphoramidite in a 5 steps processes by taking advantages of BOP mediated SNAr reaction.
In the abstract please before BOP write the compound name (benzotriazol-1-yloxytris(dimethylamino)phosphonium hexafluorophosphate (BOP))
Please insert the Figure S1A and Figure S1B in the main manuscript.
Please correct all over the manuscript where is Figure should be Scheme.
In line 38 the authors mentioned Figure S2, in my opinion the manuscript will be improved if the authors insert this figure in the main manuscript. This figure should also be improved.
In line 40 please insert some references (for example: J. Org. Chem. 2007, 72, 26, 10194–10210)
In line 47 is not the first time that BOP appears in the manuscript, please insert the name in the first time BOP appears. (1-H-benzotriazol-1-47 yloxytris(dimethylamino)phosphoniumhexafluoro-phosphate (BOP).
In Figure 1 line 70 please insert the yields and also in the description of the figure after line 43.
In the legend of Figure 2 please correct iv) is ii); ii) is iii) and c) is iv). Also insert the yields in each transformation.
In Figure 3 line 93 please insert the yields.
In the legend of Figure 4 please correct the last ii) is iii).
In my opinion, the discussion sections were poorly developed by the authors, I give some suggestion to improve the manuscript:
- The discussion of the reaction mechanism present in the SI. Also comment on the previously reported mechanism (such as Org. Chem. 2007, 72, 26, 10194–10210)
- Comment the yields of Figure 3, and also the effect of using different amines
- Comment the yields of Figure 4,
Include in the SI document copies of the 1H, 13C NMR spectra of the synthesized compounds.
The conclusion in missing.
Author Response
Issues raised by Reviewer 1:
- In the abstract please before BOP write the compound name (benzotriazol-1-yloxytris(dimethylamino)phosphonium hexafluorophosphate (BOP))
This has been addressed - see Line 14.
- Please insert the Figure S1A and Figure S1B in the main manuscript.
We have inserted Figure S1A and S1B as Scheme 1 in the main text. Please, see Line 39.
- Please correct all over the manuscript where is Figure should be Scheme.
We have renamed previous Figures 1, 2, 3 and 4 as Schemes 3, 4, 5 & 6, respectively.
- In line 38 the authors mentioned Figure S2, in my opinion the manuscript will be improved if the authors insert this figure in the main manuscript. This figure should also be improved.
We have included the previous Figure S2 as Scheme 2 in the main text – note the scheme has been improved.
- In line 40 please insert some references (for example: Org. Chem.2007, 72, 26, 10194–10210)
Thank you – this reference has been inserted as reference 11.
- In line 47 is not the first time that BOP appears in the manuscript, please insert the name in the first time BOP appears. (1-H-benzotriazol-1-47 yloxytris(dimethylamino)phosphoniumhexafluoro-phosphate (BOP).
Thank you – this has been addressed.
- In Figure 1 line 70 please insert the yields and also in the description of the figure after line 43.
Yields have been inserted in Scheme 3 and in the description after line 65.
- In the legend of Figure 2 please correct iv) is ii); ii) is iii) and c) is iv). Also insert the yields in each transformation.
Thank you – this has been addressed. Yields have been inserted into Scheme 4.
- In Figure 3 line 93 please insert the yields.
Yields have been inserted into Scheme 5.
- In the legend of Figure 4 please correct the last ii) is iii).
This issue has been addressed. Thanks.
- In my opinion, the discussion sections were poorly developed by the authors, I give some suggestion to improve the manuscript:
- The discussion of the reaction mechanism present in the SI. Also comment on the previously reported mechanism (such as Org. Chem. 2007, 72, 26, 10194–10210)
- Comment the yields of Figure 3, and also the effect of using different amines
- Comment on the yields of Figure 4,
Thanks for these useful suggestions (we have cited the useful reference) – we have modified the discussion whilst aiming to keep it concise.
- Include in the SI document copies of the 1H, 13C NMR spectra of the synthesized compounds.
PDFs of 1H and 13C spectra have been inserted into the SI.
- The conclusion is missing.
A conclusion has been included as requested.
Reviewer 2 Report
C. J. Schofield describes an improved synthetic methodology towards the phosphoramidite protected N6-methyladenosine via a BOP mediated SNAr. This new method has some interesting advantages over some previous syntheses, for instance, involves fewer steps, resulting in a higher overall yield, while it can be used towards some structural analogues. Thus, I found enough positive features coming from this work. For instance, the introduction part shows suitably the context behind the chemistry herein involved and the current state-of-art. References cited are sufficient and pertinent. The synthetic procedures are described in detailed manner, making them reproducible by others. Characterization and experimental parts seem to be Okay. Finally, I found enough novelty/originality coming from this work. For these reasons, I consider this manuscript undoubtedly meets the standards to be published in Molecules. However, I respectfully suggest only one minor modification in order to enhance the manuscript, as follows: Figures S1, S2 and S3 are called many times in the manuscript, which force readers to have both files (Article and Electronic Supplementary Material) in hands at the reading time. Thus, I recommend moving those figures from the ESI to the manuscript. This change may help not only to make easier the reading, but also to make suitable comparisons with the methods reported by others, as well as to visualize the reaction mechanism.
Author Response
Issue raised by Reviewer 2:
However, I respectfully suggest only one minor modification in order to enhance the manuscript, as follows: Figures S1, S2 and S3 are called many times in the manuscript, which force readers to have both files (Article and Electronic Supplementary Material) in hands at the reading time. Thus, I recommend moving those figures from the ESI to the manuscript.
As stated above in response to Reviewer 1, all the previous supplementary figures have now been incorporated into the main text.
Round 2
Reviewer 1 Report
The authors improve the article and it is suitable for publication.